# Learning Structure-Aware Foundational Representation of Rat Testicular Tubules Using Multiple Instance Learning

**Vedang Kshirsagar**\*              VEDANG.KSHIRSAGAR@AIRAMATRIX.COM
**Saketh Juturu**\*                SAKETH.JUTURU@AIRAMATRIX.COM
**Geetank Raipuria**              GEETANK.RAIPURIA@AIRAMATRIX.COM
**Nitin Singhal**                 NITIN.SINGHAL@AIRAMATRIX.COM
*Mumbai, India*

**Editors:** Accepted for publication at MIDL 2026

## Abstract

Testicular toxicity is a critical factor in preclinical drug safety assessment, yet automated modelling of testicular abnormalities remains largely unexplored. Unlike liver or kidney, the testis is organized into tubules that vary substantially in size and structure, making fixed-resolution patch classification ineffective. We first demonstrate that resizing tubules significantly degrades performance particularly for larger sized tubules and a Multiple Instance Learning (MIL) model offers substantial improvements. Building on this, we introduce TBA-MIL, a transformer-based aggregation model with learnable positional embeddings that encodes the structure of tubules and is pre-trained using a self-supervised Masked Instance Modelling (MIM-MIL) framework, learning tubule representations from large-scale unlabeled data. Across four tubule types, TBA-MIL with MIM-MIL outperforms state-of-the-art MIL models and establishes a strong baseline for automated testicular toxicity assessment. Additionally, we evaluate the proposed framework on an independent toxicological study and show that the predicted abnormality distributions significantly differentiate control and treated animal tissues, consistent with expert pathologists' assessment.

**Keywords:** Histopathology, Toxicologic Pathology, Testicular Toxicity, Multiple Instance Learning, Foundation Models, Self-Supervised Learning, Masked Image Modelling

## 1. Introduction

Histopathology based drug-induced tissue injury identification is a critical step in preclinical safety assessment of a potential therapeutic agent. A pathologist identifies, characterizes, and grades any observed microscopic changes on tissue samples from dosed animals. The process provides the tissue morphological data that helps the toxicologic pathologist understand and predict a drug's potential toxic effects. The assessment allows translating potential risks to the human clinical setting and informing appropriate dosing strategies and monitoring requirements.

With advancements in digital pathology and Whole Slide Images (WSI), deep learning models have shown promising results for identifying a wide variety of tissue injuries (Zingman et al., 2024; Jaume et al., 2024a; Juturu et al., 2025; Zehnder et al., 2022; Linmans et al., 2024; Pocevičiūtė et al., 2025), which assist pathologists in determining drug-induced toxicity. However, most of these works have focused primarily on Liver (Hepatic) or Kidney (Renal) toxicity, the major metabolic and excretory organs. While liver and kidney failure are immediately life-threatening, testicular toxicity is critical for often program-ending

---

\* Contributed equally

reasons including unique and irreplaceable organ function, low tolerance for risk in non-life-saving drugs and secondary impact on hormone production. This makes Testicular Toxicity findings a decisive factor in terminating a drug program early, particularly when the drug is intended for a non-life-threatening or chronic condition in a broad patient population.

Models for liver and kidney injury detection have been developed on tissue patches extracted from WSI, as the tissue injuries can be identified without requiring larger tissue context. However, in case of testes tissue, the organ consists of tubules as sub-structures which can reflect varying degrees of toxicity. Figure 1 provides sample of normal and drug affected abnormal tubules. Creating patches from the tissue does not provide sufficient context to identify the injury. Furthermore, tubules in a WSI vary significantly in size and aspect ratio, as seen in figure 1 and 7. This makes modelling tubules for identification of drug-induced toxicity non-trivial.

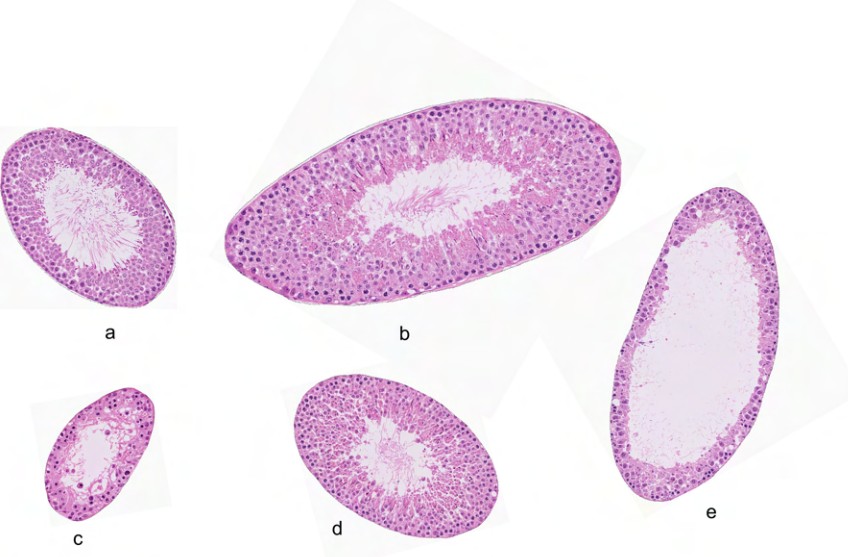

Figure 1: Sample images of testicular tubules, resized proportionally to maintain relative size. a-b) Normal tubule (No injuries, intact) c) Degeneration Tubular d) Degeneration Germ Cell e) Dilation Tubular

In this work, we first investigate the performance of state-of-the-art convolution and transformer-based feature extractors to classify testicular tubules into normal and injury classes. For this, the tubules are resized to a fixed size before passing to the model. We extensively evaluate by ablating for different input image sizes. The analysis shows that resizing tubule images significantly degrades performance on large sized tubules. Next, we model a tubule as a bag-of-words, i.e., Multiple Instance Learning, extracting features of non-overlapping patches from a tubule, followed by feature aggregation. The analysis shows that the MIL model outperforms the resized image-based classifier for all tubule sizes.

Based on this evidence, we propose a MIL model based on the transformer architecture, that incorporates positional encoding to retain the relative position of the patch instances, uses a foundational feature extractor trained on testes data, and importantly is itself pre-trained using self-supervised loss on a large scale tubular data to learn tubular representations. We compare the proposed framework with state-of-the-art MIL models

and show that it outperforms on tubular injury classification. We also perform an ablation study to show the significance of pre-training and positional embedding in the MIL model. Figure 4 provides an overview of the proposed representation learning for testicular tubules.

To the best of our knowledge, this work is the first to extensively evaluate testicular injury classification in Wistar rats. Our main contributions are as follows:

1. We demonstrate that modelling each tubule as a bag of features using Multiple Instance Learning (MIL) outperforms fixed-size image classification approaches.

2. We benchmark state-of-the-art MIL models for tubule injury classification

3. We present a foundational self-supervised pretraining strategy for tubular representation learning - Masked Instance Modelling (MIM-MIL), using a new transformer-based aggregation MIL model (TBA-MIL), that outperforms all MIL models.

4. We evaluate the utility of the tubule injury detection using the proposed approach on a toxicological study.

## 2. Related Work

### 2.1. Drug-Induced Injury Detection

Recent works on drug-induced injury detection in digital histopathology emphasize domain-specific representation learning and out-of-distribution (OOD) modelling to address the scarcity and heterogeneity of lesion annotations in toxicology pipelines. (Zingman et al., 2024) and (Dippel et al., 2024) employ supervised patch classification task to train a feature extractor, followed by detecting patches anomalous from normal tissue representation. Other works leverage self-supervised foundational models to extract features for supervised injury classification (Jaume et al., 2024a) and unsupervised anomaly detection (Juturu et al., 2025) using a neighbourhood density (K-Nearest Neighbours) in the latent space of foundation model.

Another direction of work uses generative models to learn normal tissue representation via reconstruction task, and identify anomalous tissue injury as tissue with high reconstruction error. (Zehnder et al., 2022) trained a generative adversarial network with multi-scale patches as input to enhance regional interpretation, whereas (Linmans et al., 2024) train a denoising diffusion probabilistic with a partial diffusion process to learn in-distribution image space, and found it to outperform GAN based models. Lastly, (Pocevičiūtė et al., 2025) and (Juturu et al., 2025) compared reconstruction-based approaches with methods based on latent space of foundation model. All of the above approaches work on patch level, such that each patches is classified as anomalous or a specific tissue injury type. In case of testicular abnormalities, creating tiles loses context of the tubular structure making the approach unfit for injury detection.

### 2.2. Multiple Instance Learning

Multiple-instance learning (MIL) has become the dominant computational modelling approach for weakly supervised histopathology as it allows to directly learn slide-level labels from WSI, without needing patch or pixel level labels that are expensive to obtain. A feature encoder is used to produce patch embeddings which are then pooled to produce a bag score for the entire WSI.

The encoder choice strongly affects MIL performance, a histology-tailored pretraining yields superior embeddings that are more sensitive to subtle morphological changes and less sensitive to stain variation (Wölflein et al., 2024). (Shao et al.) and (Wölflein et al., 2024) compared various publicly available foundational models for patch feature extraction

and MIL aggregation architectures, on diverse tasks, validating the utility and robustness of MIL approach for WSI feature representation learning. Both of the works show that ABMIL (Ilse et al., 2018) outperforms aggregation methods.

While effective, most widely used MIL aggregators operate on unordered sets of patch embeddings and summarize the bag into a single global representation, implicitly discarding spatial relationships and higher-order structural organization within the tissue. As a result, such approaches may be limited in their ability to capture biologically meaningful structures that depend on relative spatial arrangement.

## 2.3. Whole Slide Foundation Models

Numerous works have proposed slide-level pre-training to learn WSI representation, allowing transferability to low weakly supervised dataset regimes. These methods can be divided in to two sub-types - unimodal (Chen et al., 2022; Lazard et al., 2023; Xu et al., 2024; Lenz et al., 2025; Shao et al.) and multi-model (Jaume et al., 2024b; Shaikovski et al., 2024; Wang et al., 2024). Early works (Chen et al., 2022; Lazard et al., 2023) demonstrated that MIL trained with self-supervision can extract useful representations of WSI for down-stream tasks. (Chen et al., 2022) exploit the pyramid structure of WSIs to learn representations that capture both fine-grained morphology and coarse spatial context, by a hierarchical training using self-distillation loss. (Lazard et al., 2023) adapted self-supervised contrastive loss to gigapixel images by creating multiples representations of the same WSI. (Xu et al., 2024) adopted transformer with dilated attention (Ding et al., 2023) as slide encoder to handle massive sequence of images patches from a WSI and generate contextualized embeddings. Other methods (Wang et al., 2024; Shao et al.) employ supervised pre-training to learning aggregation layers that can be used as slide foundational models.

## 3. Dataset

The dataset used for this work consists of in-house Wistar rat testicular histopathology slides collected from pre-clinical toxicology studies. A total of 648 WSIs were available, of which 148 WSIs were used to create the supervised annotated dataset, whereas the remaining 500 were used for self-supervised training. Additionally, a toxicological study consisting of 8 Control and 16 Treated tissue WSIs is used to evaluate the effectiveness of the proposed testes tubule modelling framework. The following subsections describe the dataset preparation in detail.

## 3.1. Tubule Data Preparation

This work models testicular tubules as the fundamental building blocks of the tissue instead of using equally sized patches. To enable this, we train a segmentation U-Net model for tubule detection, which is used to extract tubule structures from the WSIs at $10\times$ magnification. Appendix section B provides details of the training setup and data used to train the tubule segmentation model. Each extracted tubule is oriented along the diagonal; that is, we consider each tubule as an ellipse and align its major axis with the diagonal of an imaginary box. This ensures geometric consistency. The region around the tubule is padded with white background using a pixel value of 250, which is similar to the background pixel values on the slide.

## 3.2. Supervised Tubular Classification Data

The supervised dataset is a collection of tubules of varying sizes, sampled and annotated by a panel pathologist from 148 WSI. It consists of a total of 10,880 tubules, assigned

to one of four classes — Normal, Degeneration Germ Cell, Degeneration Tubular, and Dilation Tubular — the latter three being common injury types. The Normal class has 5,599 samples, Degeneration Germ Cell has 1,893, Degeneration Tubular has 1,584, and Dilation Tubular has 1,804. The WSIs are split 70:30 for training and testing, 100 WSIs being used for training and 48 WSIs for testing. The training set is further split 75:25 into training and validation subsets, stratified by the number of samples in each class and tubule size. All the results are reported on test dataset.

### 3.3. Unsupervised data

From the 648 WSI, 500 are used to generate two datasets: **1) Unsupervised tubular dataset**, consisting of about 400,000 tubules including both normal and injury classes. This dataset is used to pre-train the MIL model using self-supervision to learn tubular representations; **2) Unsupervised patch dataset**, consisting of about 12 million tissue patches sampled at $5\times$, $10\times$, and $20\times$ magnification, used for training a foundational model for patch-level feature extraction.

## 4. Classifier vs Multiple Instance Learning

The initial modelling of testicular tubules is based on a naive approach to resize all tubules to a fixed size, close to the median. The resized tubules are used for training and evaluating the model. Figure 2 provides results for ConvNextV2 (Woo et al., 2023) and ViT (Dosovitskiy, 2020) models, across different model capacities. ConvNext models perform better than ViT, which can be attributed to limited availability of training data. ConvNext-Tiny performs the best achieving 90.37% balanced accuracy.

We train the best performing model (ConvNextV2-Tiny) with different input image sizes - 512, 768, 1024, and bin the results into categories based on actual tubules size, as seen in figure 3. The tubule of size greater than 1280 would be re-sized to 512 for the ConvNext-512 model, and so on. It can be observed that the performance of all models remain similar or degrade for large sized tubules. This can be due to the models' inability to model larger contextual information. Transformer models, that have better ability to build context over the entire image, are limited by the amount of training data.

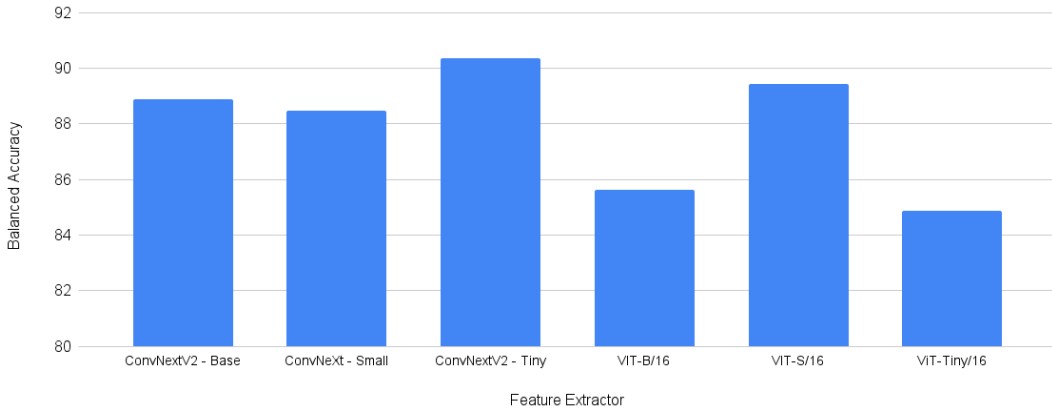

Figure 2: Performance of ConvNext(Woo et al., 2023) and ViT(Dosovitskiy, 2020) model for tubule injury classification

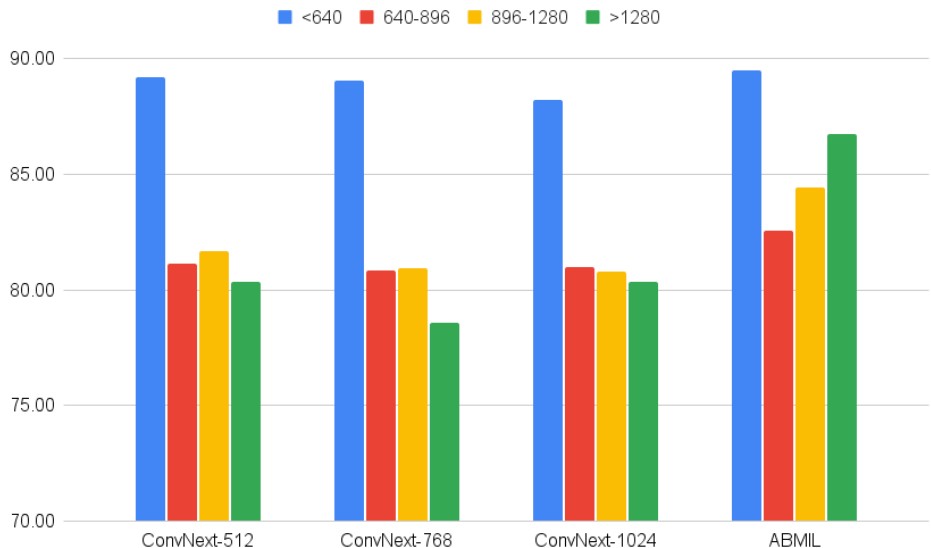

Figure 3: Analysis on performance of different model on tubules binned by their size. All convnext models are ConvNext-Tiny, and the number represents size of the input image used for training and evaluation

Since the tubule resizing approach gives poor performance on large sized tubules, a modelling approach that doesn't require resizing is needed. We hypothesize that each tubule can be modelled as a bag of patches, following the multiple instance learning (Ilse et al., 2018; Shao et al.). For this we use UNIV2 (Chen et al., 2024) as the feature extractor and ABMIL for feature aggregation, since it has shown to outperform state-of-the-art models across multiple histopathology tasks (Wölflein et al., 2024) and ability to transfer across tasks (Shao et al.).

As seen in figure 3, MIL model significantly improves the performance for all tubules sizes. We believe that this is due to a multitude of advantages 1) MIL approach does not reduce the spatial dimension of the tubule images, rather creates tiles which are passed through to feature extractor followed by feature aggregation. This would allow better attention to finer morphological features of the tubule 2) the aggregation layers using self-attention that can better extract a global context 3) better features obtained from pre-trained extractor trained on large scale histopathology data.

### 4.1. Implementation Details

All the supervised models were executed on **two NVIDIA A100 GPUs**, the use of multiple GPUs was leveraged for efficient distributed training and increased effective batch size. We experiment with learning rates [1e-3, 1e-6] depending upon the training type and models and report the best metrics accordingly. Due to the data imbalance between Normal class and other classes we used Weighted Cross-Entropy loss for better performance. We use an effective batch size 32 with gradient accumulation for all training runs, which allows training even for large image sizes. Mixed Precision was utilized for all runs. We employ the AdamW optimizer with a standard Cosine Learning Rate Scheduler. We use a mix

of geometric (rotation, flipping) and color (Brightness Contrast, HSV jitter, grayscaling) augmentations during the training phase. All results are reported as average of three runs.

## 5. Methodology

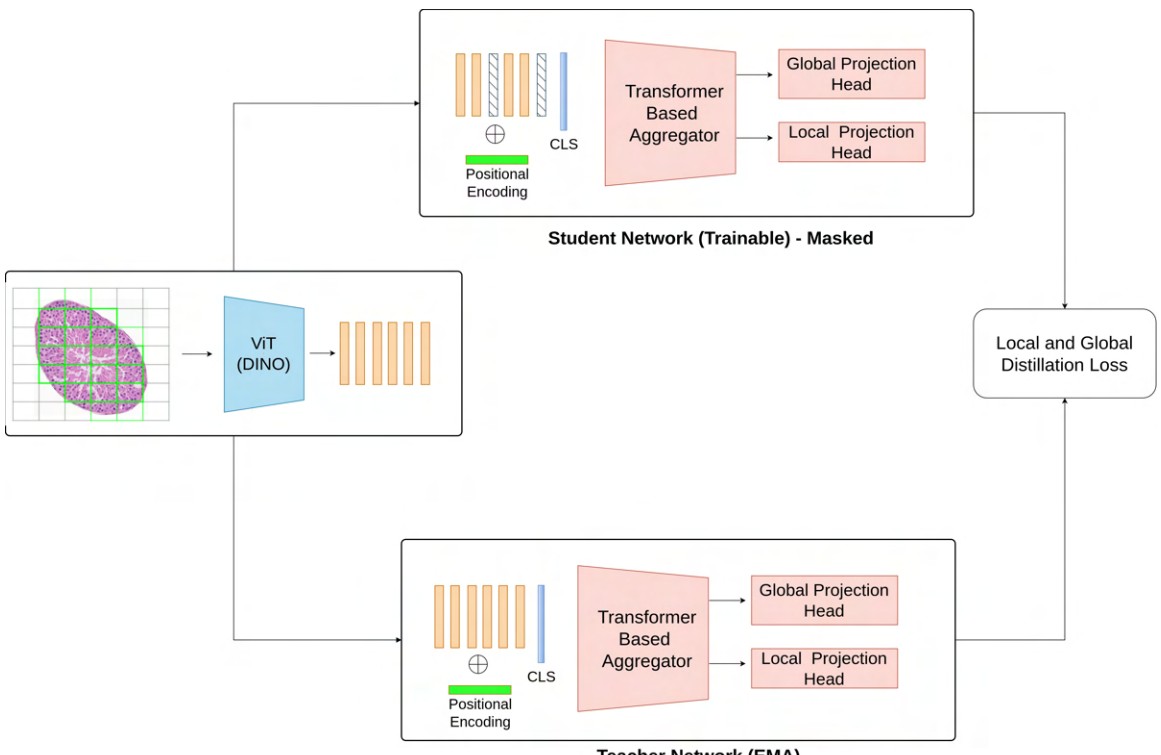

Figure 4: Overview of the proposed framework - Representation Learning for Testicular Tubules. Each tubule is decomposed into a variable-length bag of patch instances of size $224 \times 224$ and encoded using a foundational ViT feature extractor (Testes-SSL). The resulting features are aggregated using the transformer-based TBA-MIL model with learnable positional embeddings that capture the underlying tubule structure. The aggregation layers are pretrained using Masked Instance Modelling (MIM-MIL), self-supervised learning strategy, that employs student-teacher distillation to learn contextual and morphology-aware representations from large-scale unlabeled tubular data.

Based on the evidence obtained in section 4, we adopt the Multiple Instance Learning (MIL) paradigm, defining a single tubule ROI as a variable-length bag $X = \{x_1, x_2, ..., x_N\}$, where $x_i$ is a constituent patch. Our objective is to learn a system that processes this dynamic collection of patches to capture the collective structural integrity necessary for accurate abnormality classification. Figure 4 provides an overview of the proposed representation learning for testicular tubules.

## 5.1. Transformer Based Aggregation: TBA-MIL

**Feature Encoder Testes-SSL:** A Vision Transformer (ViT-Small), trained using DINO (Caron et al., 2021) self-supervised learning, extracts high-dimensional feature embeddings for each patch $x_i$.

**Structure-Aware Learnable Positional Embeddings:** To capture the essential radial context of the tubule, we incorporate learnable positional embeddings $P$. A fixed bank of embeddings $P \in \mathbb{R}^{L_{max} \times D}$ is learned, where $L_{max} = 25$ based on the largest tubule size. This learned encoding is added to the feature embedding $z_i$ of each patch before it enters the Transformer Aggregator:

$$z_i' = z_i + P_i$$

For bags shorter than $L_{max}$, only the first $N$ embeddings are utilized, imparting necessary spatial awareness.

**Patch Aggregator:** A Transformer Encoder with 4 layers, 8 attention heads and embedding dimension of size 384 is used to processes the sequence of patch features and the `[CLS]` token.

## 5.2. Masked Instance Modelling (MIM-MIL)

We propose a self-supervised framework based for pre-training MIL model using knowledge distillation and Masked Instance Modelling (MIM), inspired by previous work on patch level foundational models (Zhou et al., 2021; He et al., 2022). The setup uses two identical networks: a Student $(S)$ and a Teacher $(T)$, both sharing the same architecture as described in section. Additionally, a projection head is added to both teacher and student.

**Projection Heads:** The output features from the Aggregator are passed to two distinct, projection heads following (Zhou et al., 2021; Caron et al., 2021) that map the high-dimensional features to the prototype space $\mathbb{R}^{D_{\text{proto}}}$ (where $D_{\text{proto}}$ is the dimension of the prototypes, e.g., 8192):

- **Global Head ($h_g$):** Processes the `[CLS]` token (the bag representation) for the global loss $\mathcal{L}_{global}$.

- **Local Head ($h_l$):** Processes the patch tokens for the local loss $\mathcal{L}_{local}$.

### 5.2.1. Student-Teacher Distillation

The Teacher network $(\theta_t)$ provides stable targets for the Student $(\theta_s)$. The Teacher's weights are updated as an Exponential Moving Average (EMA) of the Student's weights, ensuring stability during training:

$$\theta_t \leftarrow \lambda \theta_t + (1 - \lambda)\theta_s$$

where $\lambda$ follows a cosine schedule, typically starting at 0.996.

## 5.3. MIM-MIL : Foundational Pre-training using Masked Instance Modelling

Our core self-supervised objective is *Masked Instance Modelling*, which forces the Student to learn the rules of tissue organization by prediction.

### 5.3.1. STOCHASTIC INSTANCE MASKING

We apply a stochastic binary mask $M \in \{0, 1\}^N$ to the bag $X$. The Student receives a corrupted view $\tilde{X}$, where the patches are masked randomly with mask ratio ranging from 0 to 0.3 and replaced by a learnable [MASK] token $e_{mask}$. The Teacher processes the original, uncorrupted bag $X$. Following standard protocols (Zhou et al., 2021; He et al., 2022), positional embeddings are added to all tokens, including the masked tokens, to ensure the model retains spatial context despite the corruption.

### 5.3.2. SEMANTIC DISTILLATION LOSS

We utilize a loss to match the student's prediction to the teacher's semantic assignment (prototypes), with the Teacher acting as an online tokenizer. We employ a cross-view strategy where the Student and Teacher process two different augmented views, $u$ and $v$, of the same image bag. The total loss ($\mathcal{L}$) combines a symmetrized Global Loss (on CLS tokens) and Local Loss (on masked patches):

$$\mathcal{L} = \frac{1}{2}(\mathcal{L}_{global}^1 + \mathcal{L}_{global}^2) + \frac{1}{2}(\mathcal{L}_{local}^1 + \mathcal{L}_{local}^2)$$

The *Global Loss* terms enforce consistency between the global representations (CLS token) of the two views. The *Local Loss* ($\mathcal{L}_{local}$) minimizes the Cross-Entropy between the Student's predicted distribution ($p_s$) for the masked view $u$ and the Teacher's sharpened distribution ($p_t$) for the clean view $u$:

$$\mathcal{L}_{local} = - \sum_{i \in \text{Masked}} p_t(u_i) \cdot \log p_s(\tilde{u}_i)$$

where $p_t$ and $p_s$ are the softmax probabilities sharpened by temperature parameters $\tau_t$ and $\tau_s$ respectively. This Cross-Entropy formulation ensures the model learns to identify structural components, optimizing the latent space for downstream classification of abnormalities.

The TBA-MIL architecture and MIM-MIL framework are closely coupled and mutually reinforcing. TBA-MIL incorporates positional encodings which are essential for stochastic instance masking (He et al., 2022); without them, the model would lack information regarding the spatial location of the masked tokens. Furthermore, TBA-MIL explicitly outputs both a global CLS token and local patch-level tokens, enabling the model to effectively learn from both global and local self-supervised distillation losses.

## 5.4. Implementation Details

The MIM-MIL pre-training was conducted on four NVIDIA A100 GPUs for 200 epochs, using the AdamW optimizer with a base learning rate of $1 \times 10^{-4}$ and a batch size of 128. We utilized a cosine decay learning rate scheduler and employed automatic mixed precision to enhance memory efficiency. Following the implementation of (Zhou et al., 2021), a stochastic masking strategy is employed with the masking ratio uniformly sampled from the range $[0, 0.3]$, for the semantic distillation loss, the student temperature $\tau_s$ is fixed at 0.1, while the teacher temperature $\tau_t$ follows a linear warm-up schedule from 0.04 to 0.07 over the first 30 epochs to ensure stable convergence.

**Supervised Fine-tuning:** The pre-trained features are used to initialize the final classification model, where the CLS token is passed through a linear layer and classified using standard supervised techniques. All results are reported as average of three runs.

| MIL Model | Feature Extractor | Pre-Training | Balanced Accuracy |
|-----------|-------------------|--------------|-------------------|
| ABMIL(Ilse et al., 2018) | UNIV2(Chen et al., 2024) | None | 91.27 |
| ABMIL(Ilse et al., 2018) | UNIV2 | Feather(Shao et al.) | 91.74 |
| DFTD(Zhang et al., 2022) | UNIV2 | None | 84.38 |
| DSMIL(Li et al., 2021) | UNIV2 | None | 91.73 |
| TransMIL(Shao et al., 2021) | UNIV2 | None | 91.71 |
| TBA-MIL | UNIV2 | None | 91.28 |
| ABMIL | Testes-SSL | None | 92.14 |
| DSMIL | Testes-SSL | None | 92.46 |
| TransMIL | Testes-SSL | None | 92.54 |
| TBA-MIL | Testes-SSL | None | 92.19 |
| **TBA-MIL** | **Testes-SSL** | **MIM-MIL** | **94.64** |

Table 1: The table compares the balanced accuracy for tubular injury classification for various MIL models, on Wistar rat test set.

## 6. Results and Discussion

We investigate the performance of state-of-the-art MIL models for supervised classification of tubular injuries, including ABMIL (Ilse et al., 2018), DFTD (Zhang et al., 2022), DSMIL (Li et al., 2021), TransMIL (Shao et al., 2021) and our proposed TBA-MIL, using UNIV2 and Testes-SSL weights. We also compare the impact of self-supervised pretraining of ABMIL model using Feather(Shao et al.) and our proposed MIM-MIL. Table 1 provides the results.

Testes-SSL consistently improves the performance of all evaluated MIL aggregators, highlighting the benefit of learning features from large-scale, domain-specific patch data. Notably, ABMIL equipped with Testes-SSL outperforms ABMIL using Feather pre-trained features, despite the latter being trained on large-scale, heterogeneous histopathology datasets. This indicates that domain-specific representation learning is particularly beneficial for testicular histopathology. When trained with Testes-SSL features, TBA-MIL achieves performance comparable to ABMIL, while TransMIL attains marginally higher accuracy, suggesting that architectural differences alone contribute limited gains in the absence of additional pre-training. In contrast, coupling TBA-MIL with MIM-MIL pre-training on large-scale unlabeled tubule data yields the strongest performance overall. This improvement is enabled by TBA-MIL's ability to model token-level representations with learnable positional embeddings, which are essential for stochastic instance masking and joint global–local distillation.

We also perform an ablation on MIM-MIL pre-training strategy to evaluate the importance of masking and positional embedding. As seen in table 2, both techniques aid in learning better feature representations. Testicular tubules have radial structure, as seen in figure 1, which can explain the utility of positional embedding as this allows the MIL model to localize the patches. On the other hand masking helps learning diverse features by compressing redundant visual patterns and thereby enforcing global tissue understanding.

Finally, we visualize the attention scores of patches within individual tubules, obtained from the TBA-MIL model pre-trained using the MIM-MIL framework on a large-scale unlabeled tubule dataset seen in figure 5. It is observed that low-information regions, such as background-dominated patches, are consistently assigned lower attention weights. In normal tubules, attention is broadly distributed across most patches, whereas in injured tubules, patches corresponding to pathological regions receive higher attention scores. These observations further provide evidence that the model learns relevant features by focusing on pathology informative regions while effectively down weighting background noise.

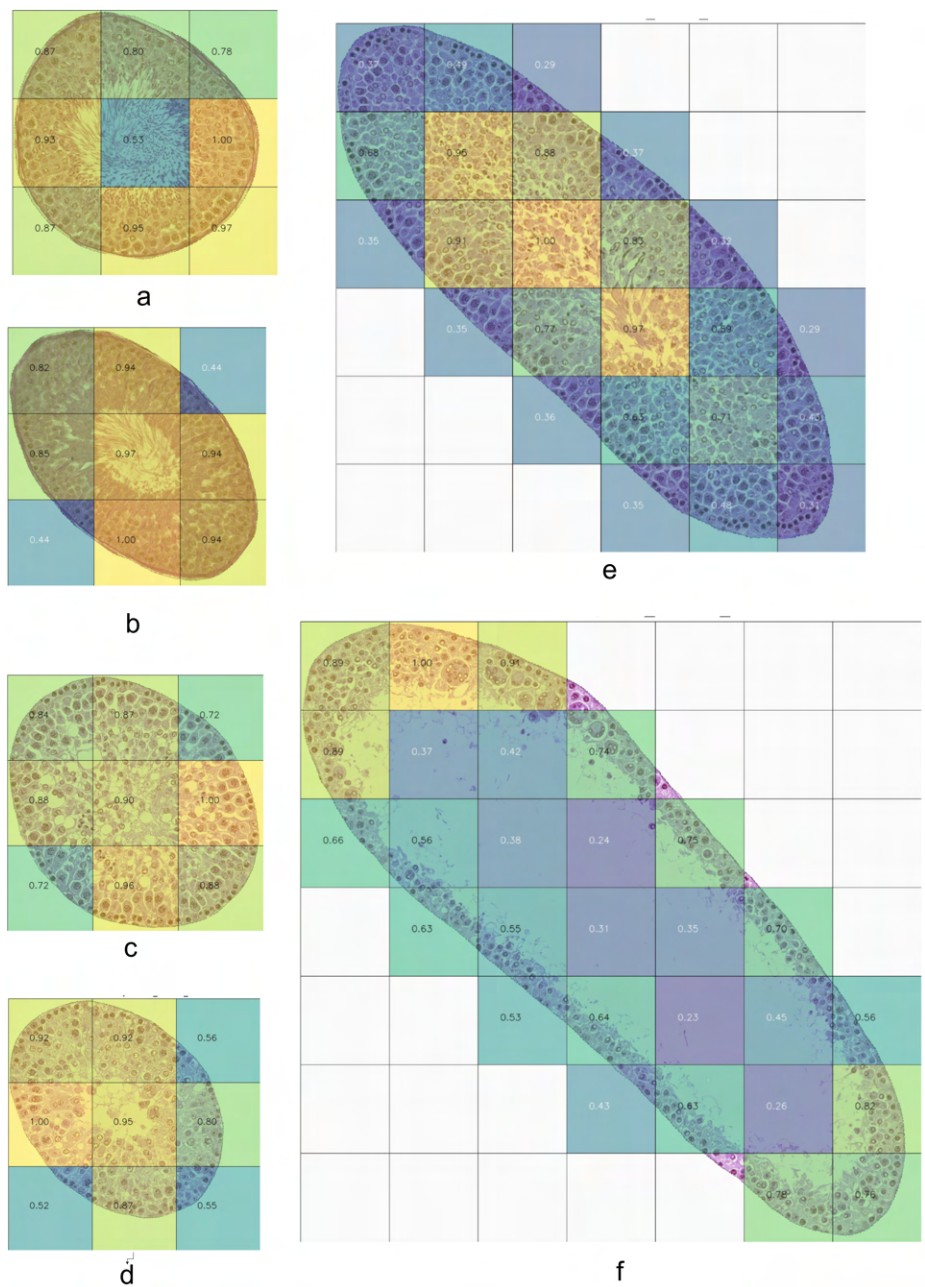

Figure 5: Patch attention scores for tubules of different classes and sizes, obtained using TBA-MIL & MIM-MIL framework. **a-b**: Normal Tubules, all patches except majority background patches get high attention scores; **c**: Degeneration Tubular, patches exhibiting signs of degeneration get higher attention; **d-e**: Degeneration Germ Cell, patches with cellular level injury obtain a high attention score; **f**: Dilation Tubular, the attention focuses on thinning of epithelium and reduced germ cell layers

| Masking | Pos Embedding | Feature Extractor | Balanced Accuracy |
|---------|---------------|-------------------|-------------------|
| Yes | No | Testes-SSL | 90.6 |
| No | Yes | Testes-SSL | 91.19 |
| **Yes** | **Yes** | **Testes-SSL** | **94.64** |

Table 2: Ablation for use of Masked Instance Modelling and Positional Embedding in MIM-MIL pre-training, the table provides balanced accuracy on Wistar rat test set.

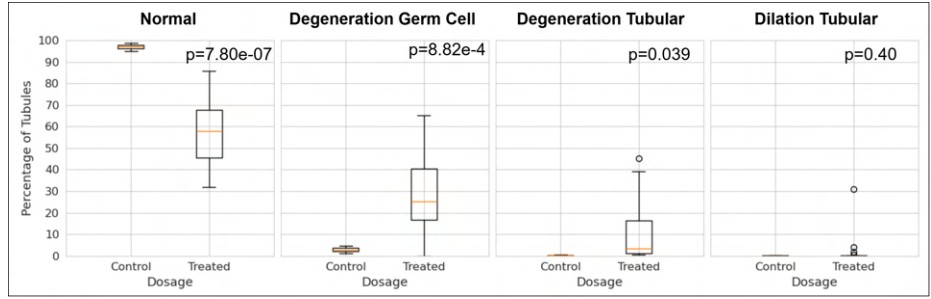

Figure 6: Evaluation of proposed framework on a Wistar Rat toxicology study, to detect testicular tubule injury, on administration of a test drug. The figure shows box plots of percentage of tubules exhibiting injury in control and drug-treated WSI.

## 6.1. Analysis on Toxicological study

We evaluate the utility of the TBA-MIL and MIM-MIL framework in assessing the toxicological effects of an administered compound. Specifically, we analyze the distribution of four tubule classes in a toxicological study by comparing the percentage of injured tubules in control and drug-treated tissues. An increased proportion of tubule injury in the treated group is indicative of compound-induced toxicity. The box plot in Figure 6 shows a statistically significant increase in the percentage of tubules exhibiting degeneration germ cell and degeneration tubular in the treated group compared to the control group. These findings are consistent with independent assessments by a panel of expert pathologists, who confirmed that the compound induces male reproductive toxicity characterized predominantly by these two injury patterns, thereby validating the proposed framework.

## 7. Conclusion

This work shows that fixed-size tubule classification is insufficient for modelling the diverse morphology of rat testicular tubules. By treating each tubule as a variable-length bag of patches, MIL provides a more effective representation that preserves spatial structure. By integrating the TBA-MIL architecture with the MIM-MIL self-supervised pre-training strategy, the proposed approach effectively exploits large-scale unlabeled data and significantly improves performance over existing MIL baselines. Attention visualizations indicate that the learned representations emphasize pathology relevant regions while suppressing background noise, and evaluation on a toxicological study demonstrates the framework's ability to detect statistically significant injury differences between control and treated groups in agreement with expert pathology assessments.

## Acknowledgments

We sincerely thank Dr. Milind Dalvi and Pranab Samanta for their valuable domain expertise in testicular tissue pathology and support in creation of the dataset.

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

## Appendix A. Sizes Distribution of Testicular Tubules

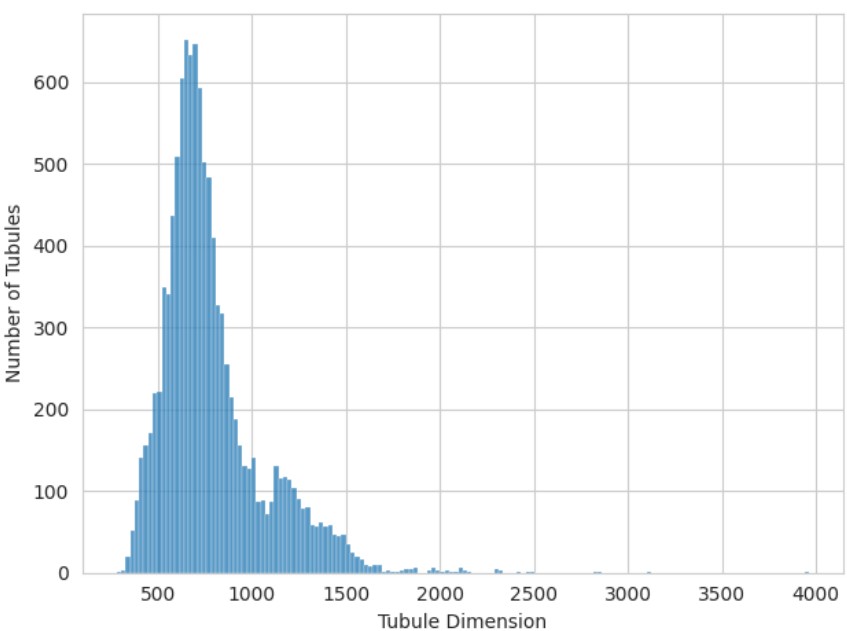

Figure 7: Histogram for tubule sizes from train and test dataset. Median tubule size is 718 pixels, at 10x magnification.

## Appendix B. Tubule Segmentation Model

Tubules form the fundamental structural and functional units of the testis and are the primary targets of testicular toxicity, with injury manifesting as structural or cellular disruptions. Each tubule can reflect varying degrees of toxicity, and therefore must be modelled individually. To this end, we develop a tubule segmentation algorithm using a U-Net architecture.

The dataset consists of approximately 12,000 overlapping patches ($1024 \times 1024$) at $10\times$ magnification extracted from 20 WSIs. These WSIs are split into training, validation and test sets (12:3:5). The WSIs used for developing the U-Net model are separate from those used to create supervised or unsupervised datasets for learning tubule representations.

The model is trained using Dice loss with a cosine decay learning rate scheduler with warm-up, a maximum learning rate of 1e-3, a batch size of 8, and a total of 100 epochs. The

model achieves a Dice score of **96.4%** on the test set. Figure 8 provides sample predictions from the tubule segmentation model. It can be observed that segmentation mask prediction is sharp even in extreme injury cases, with minor missed boundary regions in a few injury tubules.

Qualitatively we observe that the classification performance is robust to segmentation imperfections. This can be due to multitude of reason, tubules are represented as bags of patch instances rather than relying on precise pixel-level boundaries, second, the transformer-based aggregator is trained to emphasize informative patch patterns, training on large scale unsupervised tubule dataset, allowing it to down-weight noisy or less informative instances arising from minor segmentation errors, as seen in figure 5. Finally, large scale pretraining on 400,000 tubules allows the model to generalize to a variety of tubule sizes, shapes and minor error in mask.

## Appendix C. Confusion Matrix on TestSet

Figure 9 provides the confusion matrix for baseline and best performing models. It can be observed that performance gain is observed across all classes using our proposed framework; TBA-MIL pre-trained with MIM-MIL. The highest gain is obtained for classes Degeneration Germ Cell and Degeneration Tubular, reducing both false positives and false negative predictions.

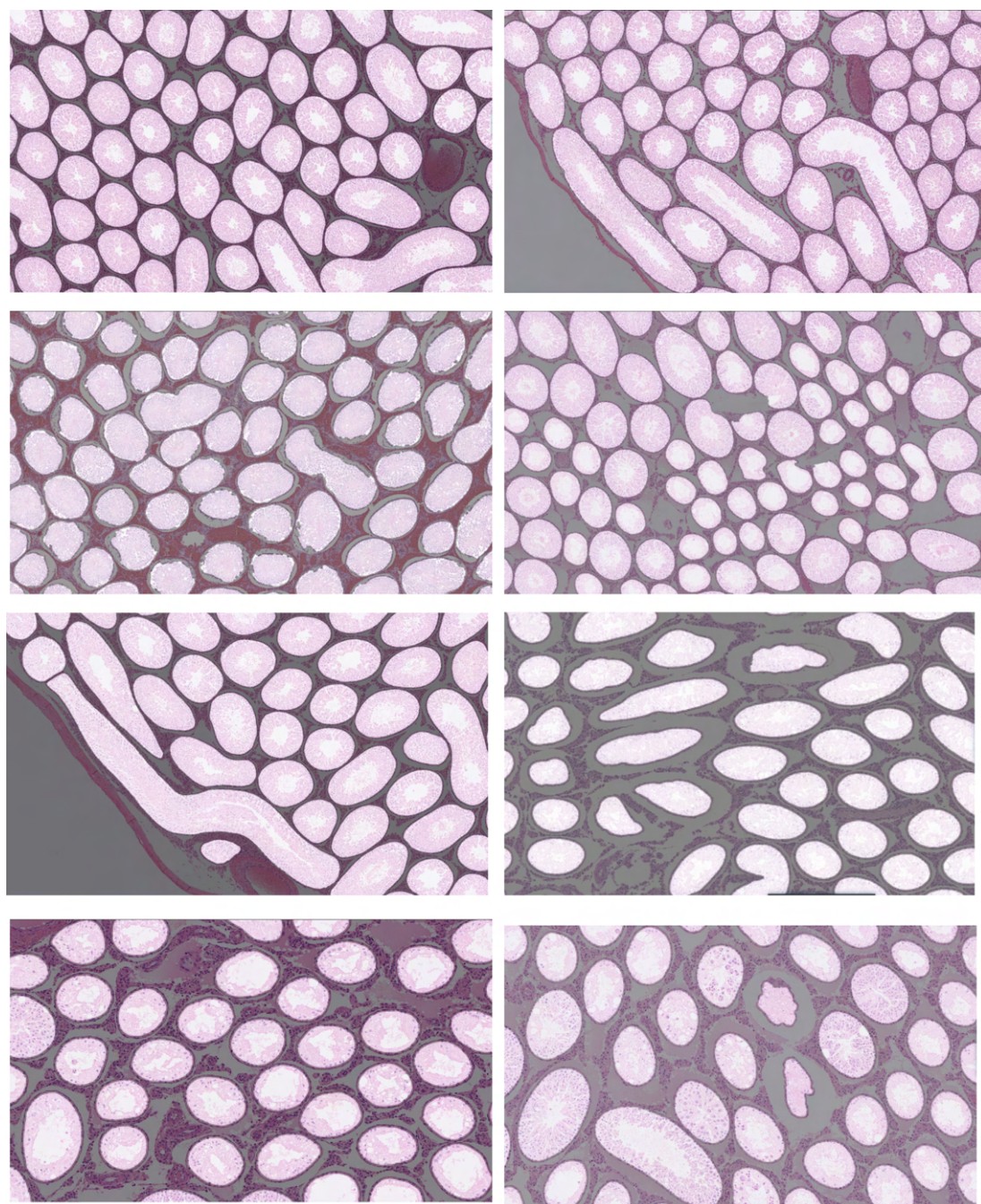

Figure 8: Sample segmentation mask predictions (overlay) on unseen data on normal and injury affected tubules. Row 1 (top): Normal Tubules, Row 2-4: Injury affected tubules.

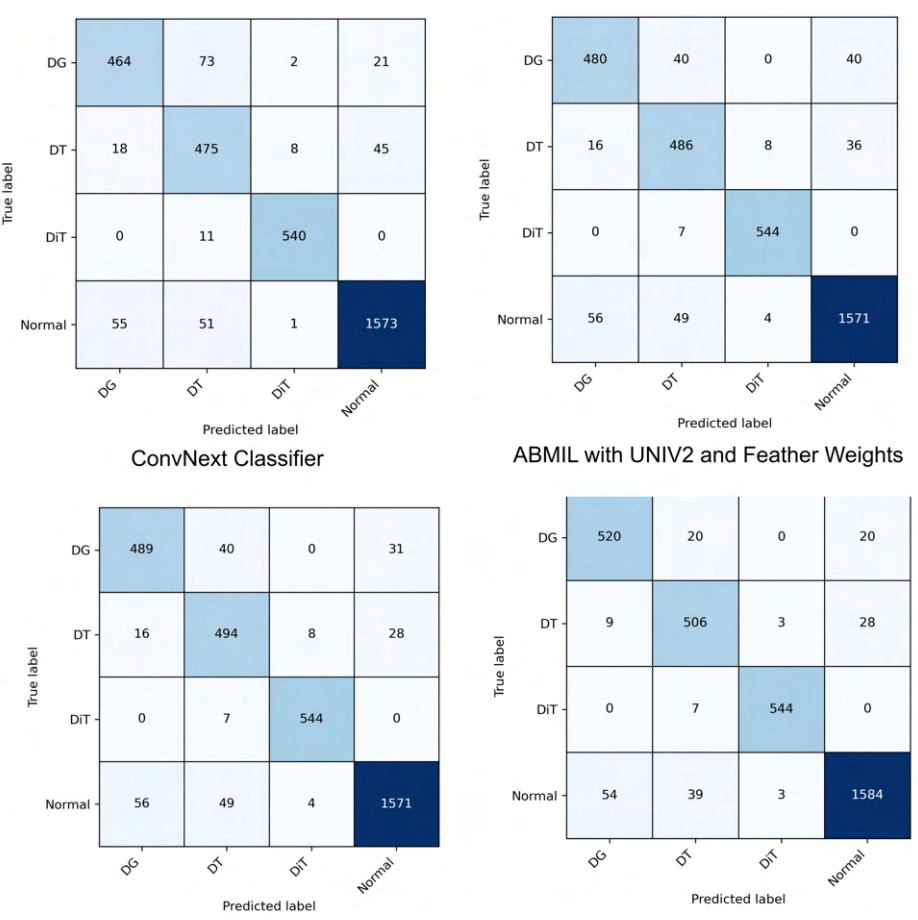

Figure 9: Confusion Matrix obtained on testset for image Re-sizing based Classifier, AB-MIL, TransMIL and TBA-MIL with MIM-MIL. Class names: Degeneration Tubular $DT$, Degeneration Germ Cell $DG$, Dilation Tubular $DiT$

