# OpenReview forum: "Learning Structure-Aware Foundational Representation of Rat Testicular Tubules Using Multiple Instance Learning"
_MIDL.io/2026/Conference — MIDL 2026 Poster_

### Official Review · Reviewer_Gdhj · 2026-01-08

**Confidence:** 4
**Preliminary Rating:** 4
**Final Rating:** 5

**Summary:**

This paper addresses the challenge of automated testicular toxicity assessment in preclinical drug safety, where traditional fixed-resolution patch classification fails due to high variability in tubule size and structure. The authors demonstrate that resizing tubule images degrades performance, particularly for larger structures, and then propose TBA-MIL (Transformer-Based Aggregation) which utilizes learnable positional embeddings to capture the unique radial context of tubules. They also introduce MIM-MIL, a self-supervised masked instance modeling framework employing student-teacher distillation for pre-training transformer aggregator inorder to  learn morphology-aware representations of testicular tissue based on large-scale unlabeled data. They utilize Wistar rats dataset,  with ~400K unlabeled tobules for MIM-MIL aggregator pre-training while >10K annotated tubules for fine-tuning the aggregator. The experiments show that the combined workflow achieves a balanced accuracy of 94.64%, significantly outperforming state-of-the-art MIL models.

**Strengths:**

- The development of the TBA-MIL and MIM-MIL framework seems to be novel and scientifically grounded technical contribution.
- The authors provide convincing evidence that traditional fixed-resolution classification fails as tubule size increases, whereas their structure-aware approach maintains high performance by treating tubules as variable-length "bags". Introduction of learnable positional embeddings is a smart addition, as it specifically captures the unique radial structure of testicular tubules, which standard MIL models ignore.
- Additionally, the foundational pre-training leveraging 12M patches for feature extraction and 400K unlabeled tubules for self-supervised learning, adds substantial value to building robust representations. The resulting 94.64% balanced accuracy is a strong indication that clearly outperforms existing state-of-the-art MIL models like TransMIL and DSMIL.
- The paper is well-structured and follows proper scientific guidelines. The authors provide clear implementation details, including hardware specs and specific hyper-parameters.

**Weaknesses:**

- The authors claim the approach establishes a strong baseline , yet the evaluation is limited to a single dataset. Testing the TBA-MIL framework on a separate, independent cohort would strengthen the claims regarding the model’s reliability and its ability to handle inter-lab staining variations or different slide scanners.
- The paper notes that TBA-MIL without the MIM-MIL pre-training (91.28%) performs similarly or slightly lower than, TransMIL (91.71%). This suggests that the architectural innovation of positional embeddings may be less critical than the large-scale pre-training phase, a point worth fully exploring more.

**Detailed Comments:**

- In Table 1, the authors compare several MIL models using the UNI2 feature extractor. However, it is not explicitly stated whether the proposed Testes-SSL feature extractor was also evaluated with the baseline MIL models (like TransMIL or DSMIL) to determine if the performance gains are primarily due to the domain-specific features or the TBA-MIL architecture.
- Figure 2 and Figure 3 illustrate performance metrics, but the captions do not explicitly state that these results are based on the Wistar rat test set. Standardizing captions to include the data source would improve the document's self-contained clarity
- Additionally, it would be valuable to specify if the Wistar rat dataset is in-house proprietary dataset or publicly available one, so as to assess reproducibility

**Justification Of Final Rating:**

I thank the authors for addressing all questions and providing significant updates to the manuscript. The additional experiments in Table 1 effectively decouple the performance gains, demonstrating that while domain-specific Testes-SSL features benefit all models, the TBA-MIL architecture provides a unique and substantial boost that baselines cannot achieve. The technical explanation regarding the incompatibility of unordered aggregators (ABMIL/TransMIL) with the MIM-MIL pre-training strategy seems to be convincing, as it justifies the necessity of a structure-aware approach for this specific geometry. Furthermore, their inclusion of an independent toxicological cohort analysis, which yielded results consistent with expert pathologist assessments, demonstrates the framework’s reliability. Given this expanded technical analysis, transparency and additional detailed justifications, I am happy to increase my rating to a Strong Accept.

**Justification Of The Preliminary Rating:**

The paper provides a practical answer to a critical challenge in drug safety assessment by developing a structure-aware framework specifically designed for the unique radial geometry of testicular tissue. The achievement of 94.64% balanced accuracy is an indication that it significantly outperforms established MIL benchmarks. However, the rating is currently a WEAK ACCEPT because the experimental results do not fully decouple the performance gains attributed to the domain-specific Testes-SSL features versus the TBA-MIL architecture and MIM-MIL pre-training. Addressing these transparency gaps in the rebuttal, particularly by benchmarking baseline models under additional identical pre-training and feature extraction conditions, would significantly strengthen the evaluation and help verify unique value of the structure-aware aggregator.

**Questions To Address In The Rebuttal:**

- In Table 1, TBA-MIL without pre-training (91.28%) appears to perform slightly below TransMIL (91.71%) and close to ABMIL. To isolate the value of your architectural contributions, could you provide a comparison where TransMIL and ABMIL is also evaluated by including the MIM-MIL pre-training strategy? (with and without Testes-SSL feature extractor).
This would help determine if the performance leap is primarily a result of the large-scale self-supervised training, domain-specific features or the specific design of the structure-aware aggregator.

---

> ### Author Response · Authors · 2026-01-24
> **Official Comment by Authors: Part**
>
> We thank the reviewer for their comprehensive summary and valuable feedback on our work. We address the weaknesses and provide clarification to the questions in detail:
>
> ```
> 1) The authors claim the approach establishes a strong baseline , yet the evaluation is limited to a single dataset. Testing the TBA-MIL framework on a separate, independent cohort would strengthen the claims regarding the model’s reliability and its ability to handle inter-lab staining variations or different slide scanners.
> ```
>
> We thank the reviewer for the comment. In the revised manuscript, we have added an additional analysis on an independent toxicological study comprising 8 control and 16 treated tissues that were not used during model training or testing (Section 6.1). While this study originates from the same data source, it represents a distinct experimental cohort and enables assessment of the framework in a realistic laboratory use case, specifically for evaluating compound-induced toxicity rather than supervised classification performance.
>
> The model successfully identified statistically significant differences between control and treated groups, and these findings were consistent with independent assessments by a panel of expert pathologists. It provides complementary evidence of the framework’s robustness and practical utility in a real toxicological setting.
>
> ```
> 2) To isolate the value of your architectural contributions, could you provide a comparison where TransMIL and ABMIL is also evaluated by including the MIM-MIL pre-training strategy? (with and without Testes-SSL feature extractor). This would help determine if the performance leap is primarily a result of the large-scale self-supervised training, domain-specific features or the specific design of the structure-aware aggregator.
> ```
> We thank the reviewer for the detailed and insightful question. To better isolate the contributions of feature pre-training and aggregation design, we have added additional comparative results in Table 1, where ABMIL and TransMIL are evaluated using the proposed Testes-SSL feature extractor. These results show that domain-specific self-supervised pre-training consistently improves the performance of all MIL aggregators, confirming the importance of large-scale patch-level representation learning.
>
> Importantly, the largest performance gains are observed when combining TBA-MIL with the proposed MIM-MIL pre-training on large-scale unlabeled tubule data. While Testes-SSL improves all models, MIM-MIL provides an additional and substantial boost that is only realizable with TBA-MIL. This is because MIM-MIL relies on learnable positional embeddings and token-level representations to support stochastic instance masking and joint global–local distillation, which are not supported by ABMIL or TransMIL. These aggregators operate on unordered sets and produce a single global representation, making them incompatible with masked token prediction and prototype-based distillation.
>
> Large-scale self-supervised pre-training is found to be the key contributor to performance improvements. We have clarified this distinction in the revised manuscript to better highlight the complementary roles of domain-specific pre-training and structure-aware aggregation.
>
> ```
> 3) Figure 2 and Figure 3 illustrate performance metrics, but the captions do not explicitly state that these results are based on the Wistar rat test set. Standardizing captions to include the data source would improve the document's self-contained clarity.
>
> ```
>
> We thank reviewer for the suggestion and have made the necessary changes in the figure captions.
>
> ```
>  4. Additionally, it would be valuable to specify if the Wistar rat dataset is in-house proprietary dataset or publicly available one, so as to assess reproducibility
> ```
> The dataset used for this work consists of in-house Wistar rat testicular histopathology slides and is not publicly available. The access to the data is subject to institutional approvals, due to proprietary constraints. The same has been reflected in the dataset section 3. of the paper.
>
> We thank the reviewer for their time in reviewing our response and for their continued engagement.

---

### Official Review · Reviewer_BAsX · 2026-01-12

**Confidence:** 3
**Preliminary Rating:** 4

**Summary:**

This paper targets rat testicular toxicity modeling, where tubules vary a lot in size/shape so “resize-to-fixed-image + classify” breaks, especially on large tubules. The authors first show this degradation empirically, then switch to a tubule-as-a-bag formulation and propose TBA-MIL, a transformer aggregator with learnable positional embeddings to retain tubule structure, paired with a testes-specific patch encoder (Testes-SSL, ViT-S trained with DINO). They further pretrain the MIL aggregator itself using MIM-MIL, a masked instance modeling + student–teacher distillation objective (global + local losses) to learn tubule-relevant contextual representations.

**Strengths:**

The paper picks a problem that’s genuinely underexplored in tox pathology (testes), and the failure mode of resizing is convincing and practically important for tubules with large context. The proposed pipeline is coherent: (1) domain-specific patch SSL (Testes-SSL), (2) structure-aware aggregation with positional embeddings, and (3) MIL-level SSL (MIM-MIL). The model choices are not flashy, but they’re well matched to the data constraints. Results are solid for a first pass at this domain: Table 1 shows a clean jump when adding Testes-SSL and then MIM-MIL (ending at 94.64 balanced accuracy), and Table 2 supports that masking + positional embedding both matter.

**Weaknesses:**

The split seems to be at the tubule level (70/30 with size distribution matched). If tubules from the same WSI/animal appear in both train and test, that’s a big leakage risk in histology. A strict WSI/animal-level split (or drug-level split) would make the claim much stronger. Baseline comparisons could be tightened: most MIL baselines use UNIV2 as encoder, while the proposed method benefits from a testes-specific SSL encoder and MIL pretraining. A fairer decomposition would test other aggregators with Testes-SSL and/or TBA-MIL with a public encoder, to isolate where the gain really comes from.

**Detailed Comments:**

Please clarify the data split unit (tubule vs WSI vs animal vs drug). If it’s tubule-level, add a WSI/animal-level test split. or MIM-MIL, you define global + local distillation (masked tokens only), but I’d like a bit more intuition on the prototype space and temperatures, why these settings are stable for variable-length bags. A small qualitative figure (attention maps over tubule patches) would help verify that the transformer is focusing on plausible injury regions instead of background padding artifacts.

**Justification Of The Preliminary Rating:**

accept because the paper convincingly shows why fixed-resolution classification is the wrong abstraction for testes tubules, and it offers a reasonable, well-motivated MIL + SSL recipe that produces a strong baseline for a niche but important tox task. The incremental gains from domain SSL and MIL pretraining are consistent and supported by ablations.

**Questions To Address In The Rebuttal:**

What happens under WSI/animal-level (and ideally drug-level) splits? Does the ~94.6% balanced accuracy hold up? How sensitive is the method to segmentation errors and to the ellipse-alignment/padding choices? Any perturbation experiment?

---

> ### Author Response · Authors · 2026-01-24
> **Official Comment by Authors: Part -1**
>
> We thank the reviewer for carefully reading our paper and for highlighting both its strengths and limitations, which helped us improve the manuscript.
>
> ```
> 1)  Please clarify the data split unit (tubule vs WSI vs animal vs drug). If it’s tubule-level, add a WSI/animal-level test split.
> ```
> We thank the reviewer for raising this concern, we would like to clarify the data split. As standard practice, the test data is created from a held out set of 48 WSI, we apologize for not explicitly highlighting this in the manuscript. The same has been added to Section 3.1.1 of the manuscript. Additionally, we have added another section (6.1) on study level analysis, to evaluate the model on control and treated WSI, both of which were held out from training and test dataset.
>
> ```
> 2)  For MIM-MIL, you define global + local distillation (masked tokens only), but I’d like a bit more intuition on the prototype space and temperatures, why these settings are stable for variable-length bags
> ```
> We thank the reviewer for this insightful question. \
> In MIM-MIL, the prototype space enables alignment between student and teacher representations across both global (CLS) and local (masked token) views. Rather than operating directly in the raw embedding space, distillation through prototypes encourages clustering of semantically similar tubule patterns while remaining agnostic to the number of instances in a bag. By projecting variable-length bags into a fixed-dimensional prototype space, the model focuses on distributional structure within each tubule.
> The global loss (CLS token) enforces consistency at the tubule level, while the local loss is applied only to masked patch tokens, encouraging the model to reconstruct meaningful local structure. This separation helps the model learn both coarse tubule-level semantics and fine-grained pathological patterns.
>
> The temperature discrepancy ($\tau_s > \tau_t$) is designed to structure the learning dynamic. The lower teacher temperature  acts as a sharpener, forcing the teacher to provide high-confidence "hard" targets, while the higher student temperature encourages smoothness and exploration. The linear warm-up of $\tau_t$ specifically prevents early instability when the teacher’s weights are still initializing, ensuring that the student is not penalized for failing to match noisy, over-confident targets early in training
>
> ```
> 3) A small qualitative figure (attention maps over tubule patches) would help verify that the transformer is focusing on plausible injury regions instead of background padding artifacts.
> ```
> We thank the reviewer for suggesting the inclusion of attention maps over tubule patches to enhance the interpretability of the model.
>
> We have added attention maps in Figure 5. It is observed that low-information regions, such as background-dominated patches, are consistently assigned lower attention weights. In normal tubules, attention is broadly distributed across most patches, whereas in injured tubules, patches corresponding to pathological regions receive higher attention scores. These observations further provide evidence that the model learns relevant features by focusing on pathology informative regions while effectively down weighting background noise.
>
> ```
> 4) Baseline comparisons could be tightened: most MIL baselines use UNIV2 as encoder, while the proposed method benefits from a testes-specific SSL encoder and MIL pretraining. A fairer decomposition would test other aggregators with Testes-SSL and/or TBA-MIL with a public encoder, to isolate where the gain really comes from.
> ```
> We again thank the reviewer for carefully reviewing the paper and for raising this insightful suggestion.
>
> To address this, we have expanded Table 1 to include results where ABMIL and TransMIL are evaluated using the Testes-SSL encoder, replacing the public UNI-V2 features. These results demonstrate that domain-specific self-supervised pre-training at the patch level provides consistent performance improvements across all MIL aggregators, confirming that part of the gain indeed comes from the testes-specific encoder.
>
> Importantly, the largest performance gains are observed when combining TBA-MIL with the proposed MIM-MIL pre-training on large-scale unlabeled tubule data. While Testes-SSL improves all models, MIM-MIL provides an additional and substantial boost that is only realizable with TBA-MIL. This is because MIM-MIL relies on learnable positional embeddings and token-level representations to support stochastic instance masking and joint global–local distillation, which are not supported by ABMIL or TransMIL. These aggregators operate on unordered sets and produce a single global representation, making them incompatible with masked token prediction and prototype-based distillation.

---

> > ### Author Response · Authors · 2026-01-24
> > **Official Comment by Authors: Part - 2**
> >
> > ```
> > 5) How sensitive is the method to segmentation errors and to the ellipse-alignment/padding choices? Any perturbation experiment?
> > ```
> > We thank the reviewer for the question.
> >
> > **Sensitivity to segmentation errors**
> > We have added details on U-Net training, evaluation metrics (96.4% Dice) in Appendix section B, as well as sample predictions from the tubule segmentation model in Figure 8. It can be observed that segmentation mask prediction is sharp even in extreme injury cases, with minor missed boundary regions in a few injury tubules.
> >
> > Furthermore, qualitatively we observe that the classification performance is robust to  imperfections by the segmentation model. This can be due to multitude of reason, first, tubules are represented as bags of patch instances rather than relying on precise pixel-level boundaries, second, the transformer-based aggregator is trained to emphasize informative patch patterns, training on large scale unsupervised  tubule dataset, allowing it to down-weight noisy or less informative instances arising from minor segmentation errors, as seen in figure 5. Finally, large-scale pretraining on ~400,000 tubules allows the model to generalize to a variety of tubule sizes, shapes and minor error in mask.
> >
> > **Sensitivity to padding strategy & ellipse alignment**
> > Ellipse alignment allows standardizing each tubule to a square shaped image after padding, which is of relevance when training baseline classifier using resizing strategy, however, since the MIL method creates tiles from the tubule, the alignment is not consequential.
> > Padding with a uniform near-background value is employed solely to standardize tensor shapes and does not introduce semantic content. Importantly, large-scale self-supervised pre-training is performed using the same padding strategy as that used during supervised training and evaluation. As a result, the model would learn to treat padded regions as low-information backgrounds, as seen in attention maps (Figure 5).  We expect the framework to be largely insensitive to the specific padding choice as long as it is applied consistently throughout training and inference. However, changing the padding strategy between training and testing may negatively impact model performance.
> >
> > We again thank the reviewer for the careful and thoughtful review of our paper, and for taking the time to read through our response.

---

### Official Review · Reviewer_JxKa · 2026-01-15

**Confidence:** 4
**Preliminary Rating:** 3

**Summary:**

This paper focuses on the structural-aware foundation representation learning of rat seminiferous tubules. Aiming at the insufficient automation modeling in preclinical drug safety assessment of testicular toxicity, it points out that traditional fixed-resolution patch classification performs poorly due to large variations in tubule size and structure, while multi-instance learning (MIL) models show superior performance. Subsequently, a Transformer-based aggregation model named TBA-MIL, which incorporates learnable positional embeddings to encode tubule structure. And a self-supervised masked instance modeling framework called MIM-MIL were designed. In the classification task of four types of tubules, TBA-MIL combined with MIM-MIL outperforms existing state-of-the-art MIL models, establishing a strong baseline for automated testicular toxicity assessment.

**Strengths:**

1. Testicular toxicity is a critical but understudied area in preclinical drug safety assessment, with few automated modeling efforts to date. This work fills a significant gap by focusing on rat testicular tubule classification, addressing a key pain point in drug development (i.e., early termination due to unforeseen testicular toxicity)。
2. The study thoroughly justifies the adoption of Multiple Instance Learning (MIL) by demonstrating the limitations of fixed-size image classification for variable-sized tubules. The proposed TBA-MIL (transformer-based aggregation with learnable positional embeddings) and MIM-MIL (self-supervised masked instance modeling) address core challenges: preserving spatial structure of tubules and leveraging large-scale unlabeled data.
3. The use of large, well-curated datasets (648 WSIs, 10k+ labeled tubules, 400k unlabeled tubules) strengthens the reliability of results.

**Weaknesses:**

1. The authors mention using a U-Net for tubule segmentation but provide minimal details on the segmentation model’s training (e.g., dataset used for segmentation, loss function, performance metrics like Dice coefficient). Segmentation errors could propagate to downstream classification, so transparency on this critical preprocessing step is lacking.
2. While the authors show MIL outperforms fixed-size classifiers, they do not explore other variable-size handling approaches (e.g., adaptive pooling, hierarchical vision transformers, or dynamic receptive field models). A brief comparison to these alternatives would strengthen the case for MIL as the optimal paradigm.
3. The authors state the mask ratio ranges from 0 to 0.3 but do not explain why this range was selected. Additionally, temperature parameters (τₜ, τₛ) for the semantic distillation loss are not reported, nor is there a sensitivity analysis to show how these hyperparameters impact performance.

**Detailed Comments:**

No further questions, thanks.

**Justification Of The Preliminary Rating:**

This submission addresses a critical yet understudied problem in preclinical drug safety assessment—testicular toxicity—through a well-motivated methodological framework (TBA-MIL + MIM-MIL) that outperforms state-of-the-art MIL models, achieving 94.64% balanced accuracy. The rigorous experimental validation, including fixed-size versus MIL comparisons and comprehensive ablation studies, combined with a large, carefully curated dataset (648 WSIs, over 10,000 labeled tubules), substantially strengthens its core contributions.

However, the paper exhibits notable limitations in several key areas. First, the generalization analysis remains limited, as the study is restricted to Wistar rat tissue with only four tubule classes. Second, critical preprocessing steps, particularly the tubule segmentation pipeline, lack sufficient methodological detail. Third, the absence of qualitative insights, such as attention map visualizations and systematic misclassification analysis, weakens interpretability. Finally, the paper provides incomplete reporting of computational efficiency metrics and inadequate justification for hyperparameter selections.

These gaps collectively hinder reproducibility and diminish practical utility, preventing a stronger rating. With targeted revisions addressing these limitations, this work has the potential to meet MIDL standards for both impact and methodological rigor.

**Questions To Address In The Rebuttal:**

1. Add a short subsection (or supplement) detailing the U-Net segmentation setup: training data (e.g., number of annotated tubules for segmentation), loss function (e.g., Dice loss, cross-entropy), and performance metrics (Dice coefficient, IoU) on a held-out segmentation test set. This ensures reproducibility and addresses potential concerns about segmentation-induced bias.
2. Given the study is restricted to Wistar rat testicular tissue and four specific tubule classes, what evidence or reasoning supports the potential generalization of the proposed TBA-MIL/MIM-MIL framework to other rat strains, human testicular tissue, or additional tubule injury types (e.g., inflammation, fibrosis)? Are there inherent biological or morphological similarities/differences that would impact transferability?
3. The U-Net segmentation model is critical for tubule extraction, but details on its training are lacking. Please specify: (1) the dataset used to train the segmentation model (e.g., number of manually annotated tubules, source WSI), (2) the loss function employed (e.g., Dice loss, cross-entropy), (3) performance metrics on a held-out segmentation test set (e.g., Dice coefficient, IoU), and (4) how segmentation errors were mitigated to avoid propagating bias to downstream classification.
4. The Testes-SSL feature extractor is a ViT-Small trained with DINO. Why was ViT-Small selected over other vision transformers (e.g., ViT-Base, ConvNextV2, UNI, CONCH, other pretrained models) for feature extraction? How does DINO pretraining on testicular data compare to other self-supervised pretraining methods (e.g., MoCo, SimCLR) for this specific task?

---

> ### Author Response · Authors · 2026-01-24
> **Official Comment by Authors : Part 1**
>
> We thank the reviewer for the constructive feedback and for highlighting both the strengths of our work and areas for improvement. Below, we address each concern in detail and describe the corresponding revisions made to the manuscript.
> ```
> 1. Details of the U-Net segmentation setup
> ```
> Thank you for highlighting this shortcoming in the manuscript. We have added the details in the Appendix Section B, including qualitative results.
>
> Summary of the U-Net segmentation setup \
> Training data : ~12,000 overlapping image patches (1024 × 1024 pixels) at 10× magnification extracted from 20 whole-slide images (WSIs) that were not included in either the supervised or unsupervised training datasets. \
> Loss Function: Dice Loss \
> Performance Metrics on Test Set: 96.4% Dice Score
>
> ```
> 2. Question : Given the study is restricted to Wistar rat testicular tissue and four specific tubule classes, what evidence or reasoning supports the potential generalization of the proposed TBA-MIL/MIM-MIL framework to other rat strains, human testicular tissue, or additional tubule injury types (e.g., inflammation, fibrosis)? Are there inherent biological or morphological similarities/differences that would impact transferability?
> ```
> We thank the reviewer for raising this point, which allows us to clarify the transferability of the proposed approach.
>
> **Generalization across different Strains of Rat** \
> Testicular tubule architecture is similar across common laboratory rat strains (e.g., Wistar, Sprague–Dawley),  morphological elements tubular organization, germ cell layering and luminal configuration are largely invariant with strain-dependent differences primarily showing up as change in injury  frequency and severity [1,2]. Because the framework learns structure-aware representations of tubules, rather than relying on strain-specific appearance features, we anticipate that the model will generalize across rat strains with minimal or no additional fine-tuning.
>
> **Extension to human testicular tissue.** \
> While the testicular representation exhibit  inter-species differences, human and rodent tubules share strong morphological similarity at the architectural level including a basement membrane, radially organized germ cell layers, central lumen formation and injury characterized by loss of germ cells, epithelial thinning, or luminal collapse, which is why rodents are part of pre-clinical toxicology assessment [3].  However, tissue extraction process, tissue morphology and the tissue size would significantly alter the tissue representation, thus requiring an independent effort to model human testicular tissue, though the proposed approach would act as a strong contender.
>
> **Applicability to additional tubule injury types** \
> The framework is inherently class-agnostic and not tailored to specific lesion definitions, injuries impact tubules through structural and cellular density alterations which are precisely the signals captured by the proposed structure-aware representation learning approach. Therefore, additional injury types can be incorporated by extending the downstream labeling without modifying the core representation learning strategy.
>
> [1] Wallig, Matthew A., et al., eds. Fundamentals of toxicologic pathology. Academic press, 2017.
> [2] Wilkinson, J. M., S. Halley, and P. A. Towers. "Comparison of male reproductive parameters in three rat strains: Dark Agouti, Sprague-Dawley and Wistar." Laboratory animals 34.1 (2000): 70-75.
> [3] Treuting, Piper M., Suzanne M. Dintzis, and Kathleen S. Montine, eds. Comparative anatomy and histology: a mouse, rat, and human atlas. Academic Press, 2017.

---

> > ### Author Response · Authors · 2026-01-24
> > **Official Comment by Authors : Part 2**
> >
> > ```
> > 3. how segmentation errors were mitigated to avoid propagating bias to downstream classification.
> > ```
> > We thank the reviewer for this insightful question on sensitivity to segmentation errors. \
> > We have added detailed information on the tubule segmentation model in Appendix Section B, including training details, evaluation metrics, and qualitative examples. The U-Net model achieves a Dice score of 96.4% on the held-out test set, and Figure 8 shows representative segmentation outputs. The predicted masks were found generally sharp and accurate, given the high dice score, including in severe injury cases, with only minor boundary inaccuracies observed in a small subset of tubules.\\
> >
> > Importantly, we observe that downstream classification performance is robust to such minor segmentation imperfections. This robustness arises from several design choices. First, tubules are represented as bags of patch-level instances rather than relying on precise pixel-level boundaries, reducing sensitivity to small boundary errors. Second, the transformer-based MIL aggregator is trained to emphasize informative patch patterns while down-weighting noisy or less informative instances. This effect is further strengthened by large-scale unsupervised pre-training on tubule data, which encourages the model to focus on pathology-relevant regions and ignore spurious artifacts, as illustrated by the attention visualizations in Figure 5. Finally, pre-training on approximately 400,000 tubules exposes the model to substantial variability in tubule size, shape, and segmentation quality, improving generalization and reducing the risk of systematic bias propagation from segmentation errors.\\
> >
> > ```
> > 4. The Testes-SSL feature extractor is a ViT-Small trained with DINO. Why was ViT-Small selected over other vision transformers (e.g., ViT-Base, ConvNextV2, UNI, CONCH, other pretrained models) for feature extraction? How does DINO pretraining on testicular data compare to other self-supervised pretraining methods (e.g., MoCo, SimCLR) for this specific task?
> > ```
> >
> > **Choice of ViT-Small backbone**
> > There were two primary motivations for choosing ViT-Small as the feature extractor. First, the scale of patch-level data used for self-supervised training is large (~12 million patches). Second, ViT-Small offers a practical trade-off between representational capacity, computational efficiency, and scalability. Larger ViT variants substantially increase GPU memory consumption and inference latency, both during feature extractor training and during downstream integration with the MIL pipeline
> >
> > **Choice of DINO pretraining**
> > DINO pre-trained has been shown to outperform all other self-supervised training [4, 5, 6] including MoCo, SimCLR, and models trained using DINO training strategy have been widely used by state-of-the-art work [4,7,8] which was the motivation for selecting DINO.
> >
> > [4] Chen, Richard J., et al. "Towards a general-purpose foundation model for computational pathology." Nature medicine 30.3 (2024): 850-862.
> >
> > [5] Caron, Mathilde, et al. "Emerging properties in self-supervised vision transformers." Proceedings of the IEEE/CVF international conference on computer vision. 2021.
> >
> > [6] Oquab, Maxime, et al. "Dinov2: Learning robust visual features without supervision." arXiv preprint arXiv:2304.07193 (2023).
> >
> > [7] Wölflein, Georg, et al. "Benchmarking pathol@ogy feature extractors for whole slide image classification." arXiv preprint arXiv:2311.11772 (2023).
> >
> > [8] Shao, Daniel, et al. "Do Multiple Instance Learning Models Transfer?." Forty-second International Conference on Machine Learning.
> >
> > ```
> > 5. The authors state the mask ratio ranges from 0 to 0.3 but do not explain why this range was selected. Additionally, temperature parameters (τₜ, τₛ) for the semantic distillation loss are not reported, nor is there a sensitivity analysis to show how these hyperparameters impact performance.
> > ```
> >
> > The hyperparameters including masking ratio and temperature parameters have been adopted from [9], this has been added to paper.
> >
> > [9 ]Zhou, Jinghao, et al. "ibot: Image bert pre-training with online tokenizer." arXiv preprint arXiv:2111.07832 (2021)
> >
> > We thank the reviewer for taking the time to review our response and for their continued engagement.

---

> ### Comment · Reviewer_JxKa · 2026-02-01
> **Official Comment of Submission286 by Reviewer JxKa**
>
> Thank you for addressing my questions thoroughly. The authors have thoughtfully resolved most of my concerns. Given the substantial improvements and the authors' attentive engagement with feedback, I am raising my score to 4.

---

> > ### Author Response · Authors · 2026-02-02
> >
> > We sincerely thank the reviewer for the time and effort invested in reviewing our manuscript. We are very encouraged to see that our revisions addressed the raised concerns and that the reviewer has decided to raise the evaluation to a 4.
> >
> > We kindly request the reviewer to update the 'Final Rating' to ensure the numerical value is updated in the system.

---

### Author Rebuttal · Authors · 2026-01-24

**Rebuttal:**

We thank the reviewers for their thorough and constructive feedback. The suggestions have helped improve the clarity and presentation of the manuscript, and we have addressed them in the responses and the **revised manuscript (attached)**.

Summary of updates to the manuscript
* Added detailed descriptions of the tubule segmentation model in Appendix Section B, including the dataset, training hyperparameters, evaluation metrics, and qualitative results.
* Expanded the Dataset (Section 3) and Methodology (Section 5) sections with additional details and clarifications.
*  Added further comparative results in Table 1 to better elucidate the sources of the observed performance gains.
* Included attention heatmaps over tubules of different classes and sizes in Figure 5 to enhance interpretability.
* Added a confusion matrix reporting performance on the test set in Appendix Section C.
* Added an analysis of an independent toxicological study comprising 8 control and 16 treated tissues in Section 6.1.

All changes in the manuscript are highlighted in red.

**Supporting Material:**

/attachment/d57679ec8f41570ce434887bacdd0899eec1d597.pdf

---

### Comment · Area_Chair_9fZX · 2026-01-31
**Request for Final Rating Update After Rebuttal**

Dear Reviewer,

This is a gentle reminder to please update or confirm your feedback and rating based on the authors’ rebuttal, as we move toward finalizing recommendations.

Thank you for your time.

---

### Meta-Review · Area_Chair_9fZX · 2026-02-07

**Recommendation:** Accept (Poster)
**Confidence:** 4

**Metareview:**

All reviewers found the proposed method to be novel and the results promising.

---

### Decision · Program_Chairs · 2026-02-14

Accept (Poster)